# Multidimensional impacts of coronavirus pandemic in adolescents in Pakistan: A cross sectional research

**Nazish Imran** [1] *, **Fauzia Naz** [2], **Muhammad Imran Sharif** [1], **Sumbul Liaqat** [1], **Musarrat Riaz** [3], **Abida Khawar** [4], **Muhammad Waqar Azeem** [5]

**1** Department of Child and Family Psychiatry, King Edward Medical University/Mayo Hospital, Lahore, Pakistan, **2** Government Islamia Graduate College, Lahore, Pakistan, **3** Baqai Institute of Diabetology & Endocrinology, Baqai Medical University, Karachi, Pakistan, **4** ZA School system, Lahore, Pakistan, **5** Department of Psychiatry, Sidra Medicine, Weill Cornell Medicine, Doha, Qatar

* nazishimrandr@gmail.com

**Data Availability Statement:** All relevant data are within the paper and its Supporting information files.

## Abstract

### Background

COVID-19 has posed unique challenges for adolescents in different dimensions of their life including education, home and social life, mental and physical health. Whether the impact is positive or negative, its significance on the overall shaping of adolescents' lives cannot be overlooked. The aim of the present study was to explore impacts of the pandemic on the adolescents' everyday lives in Pakistan.

### Methods

Following ethical approval, this cross-sectional study was conducted through September to December, 2020 via an online survey on 842 adolescents with the mean age of 17.14 ± $SD$ 1.48. Socio-demographic data and Epidemic Pandemic Impact Inventory-Adolescent Adaptation (EPII-A) was used to assess the multi-dimensional effects of the pandemic.

### Results

Among the 842 participants, 84% were girls. Education emerged as the most negatively affected Pandemic domain (41.6–64.3%). Most of the adolescents (62.0–65.8%) had reported changes in responsibilities at home including increased time spent in helping family members. Besides, increase in workload of participants and their parents was prominent (41.8% & 47.6%). Social activities were mostly halted for approximately half (41–51%) of the participants. Increased screen time, decreased physical activity and sedentary lifestyle were reported by 52.7%, 46.3% and 40.7% respectively. 22.2–62.4% of the adolescents had a direct experience with quarantine, while 15.7% experienced death of a close friend or relative. Positive changes in their lives were endorsed by 30.5–62.4% respondents. Being male and older adolescents had significant association with negative impact across most domains ($p<0.05$).

**Funding:** The author(s) received no funding for this work.

**Competing interests:** The authors have declared that no competing interests exist.

## Conclusions

Results have shown that *COVID-19* exert significant multidimensional impacts on the physical, psycho-social, and home related domains of adolescents that are certainly more than what the previous researches has suggested.

## Background

COVID-19 pandemic has brought a "parallel pandemic" of psychosocial problems hidden in the cloak. While all ages have been severely affected due to COVID-19, the psychosocial deprivation of adolescents and children cannot be neglected [1]. Adolescence is a very sensitive period with developmental growth, emotional maturity, character building and modifications in different personality dimensions. This is the period when grooming of peer affiliations grows and glows, and foundations of new possibilities including employment, choice-based higher education sprout. Coronavirus pandemic has emerged as a devastating challenge which has taken aback the societal healthcare systems and have shuddered the economic as well as social structure globally [2]. Likewise, the pandemic hit Pakistan and almost 655,000 cases (the number is continuously increasing) with over 14000 deaths are reported [3]. Along with all other cautionary measures, lockdown has also been imposed in Pakistan and the government had to close educational institutions along with many workplaces [4, 5]. This has resulted in adolescents spending most of their time at home, restricted in one environment. Advise to maintain social distancing have influenced adolescents physical, emotional, economical state of affairs in one way or the other. The influence of shutting down of workplaces adds up for adolescents either if the parents' employment is also affected or if they or their parents have to work in an exposed environment. As a result, access to food, clean water, and other basic life necessities like transportation and medications has been limited for general masses [6]. The closing of schools has led to an increased burden of virtual learning, which is a new thing for most school-going students [7, 8]. Meanwhile, all of their co-curricular, extra-curricular, athletic, and community activities have been put on a halt. Consequently, most adolescents have been spending all their time staying at home. Adolescents have to spend more time doing household chores, taking care of siblings or using gadgets e.g., mobile phones, laptops etc. This had in one way led to increased interactions within the family members and conflicting interpersonal communication with parents, siblings and/or other family members. Studies have noted increased parent-child and sibling conflicts during this pandemic [7–9]. Adolescents also feel a lack of privacy due to increased hours spent with family members [1]. On the other end of the spectrum, social restrictions have made it difficult for adolescents to maintain their social life which is important for their personal growth. They have been restricted meeting with friend, romantic partners, or any other family members who are away from their homes. Social distancing enforced people to cancel the celebration of important life events like birthdays, or graduation ceremonies that hold special meanings for children and adolescents have caused social distress [10]. Adolescents have been relying on social networking sites to compensate for the lack of social communications during this pandemic which has not only led to increased screen time throughout the day but also has disturbed the night sleep patterns which are also recognized by the contemporary researches [10, 11]. Literature also suggests the increasing frequency of sleep difficulties, stress, anxiety, and substance use in adolescents during this time [12, 13]. A sedentary lifestyle including lack of exercise and increased intake of unhealthy foods has been a major concern in the COVID-19 pandemic [12, 14]. A very

particular population of adolescents at risk of psychosocial problems is the one who has been directly affected by COVID-19 [15]. Adolescents who were exposed or infected by COVID-19 themselves or had a family member exposed or infected leading to hospitalization or quarantine, experienced depression and anxiety [4, 5, 8, 9].

In reverse, this pandemic has brought some positive changes in some adolescents as well. For example, increased time spent at home has brought family members closer. Adolescents have been able to avail free time to pursue new hobbies, constructive activities which may help in their personality growth [1]. The spirit of volunteering and donating to society is contributing to developing altruistic sense among the adolescents. They have also been able to value ordinary day-to-day things more and appreciate school or work more. This has shown to increase efficiency and productivity in their schoolwork as well as employments [1].

The multidimensional impact of COVID-19 on adolescents cannot be overlooked. Several studies across the globe have indicated that mostly developed countries have shown increase psychological as well as social morbidity in children and adolescents during this pandemic. However, to this date, to our knowledge, no study has taken multidimensional approach on different aspects of the impact of COVID-19 collectively by using *Epidemic Pandemic Impact Inventory-Adolescent Adaptation (EPII-A)* [16]. *EPII-A* is a newly developed inventory and is in the process of determining psychometric properties. The present study is one of the pioneer studies which will contribute a rich data to explore frequencies and percentages of the responses of different sub-scales of this inventory. Besides, the present study, using a multifaceted approach, aimed to explore different dimensions of the impact of COVID-19 in adolescents according to EPII-A including work and employment, educational and social activities, domestic life, economic difficulties, emotional health, physical health, social distancing and quarantine, COVID-19 infection, as well as positive changes encountered in adolescents in Lahore, Pakistan. The current study also explored likely age and gender differences, rural-urban differences regarding work and employment, educational and social activities, domestic life, economic difficulties, emotional health, physical health, social distancing and quarantine. Further, impact with presence of medical and psychiatric illness were also explored.

## Methods

### Setting and participants

After getting approval from the Institutional Review Board, King Edward Medical University, Lahore, Pakistan, the present study, by using a cross sectional research design, recruited adolescents (N = 842; females = 707 (84%); males = 135 (16%)) from different public higher secondary schools and colleges. Their age was ranged from 13 to 18 years ($Mean_{age}$ = 17.98 (SD ± 1.48). The students and their parents were approached through institutional head and only those students were recruited whose parents gave written *informed consent* to take part in the study. Both the parents and the students were provided with the objectives of the research and *participant's information sheet* related to the research. The participant's information sheet described the objectives of the research in detail. After taking written informed consent from parents and assent from adolescents, the participants were provided with the link of an online (approx. 25 minutes) google form which was to be filled by the participants. They were told that participation in the study was exclusively voluntary and if they feel uncomfortable, they can quit at any stage. They were assured that their identification will be anonymous and their information will not be disclosed to anyone and they have to submit the filled form only once. They were also briefed that the results will be calculated by taking average values, frequencies and percentages. The study was conducted during the period of September -December 2020.

## Assessment measures

**Sociodemographic information** included questions related to age, gender, place of residence (urban, rural), education level, family system, parent's education level etc. In addition, respondent's history of chronic medical problem and psychiatric illness was also collected.

   **Epidemic Pandemic Impact Inventory-Adolescent Adaptation (EPII-A),** a newly developed tool was used (after permission from authors) to assess the impact of the coronavirus disease pandemic on various domains of personal and family life relevant to adolescents [16]. EPII-A is 114 items inventory of pandemic-related experiences across ten major life domains i.e., work and employment (11-items), education and training (9-items), home life (10-items), social activities (16-items), economics (5-items), emotional health and well-being (10-items), physical health problems (12-items), physical distance and quarantine (7-items), infection history (13-items), positive change (20-items). Each item has response options of No or Yes or Not Applicable. Main psychometric results of the EPII-Adolescents Adaptation version i.e., construct validity, EFA, item analysis and Cronbach's alpha reliabilities in our population revealed good properties and are being reported separately.

## Statistical analysis

The statistical package for social science 20.0 (SPSS 20.0) program was used for all statistical analysis. First, descriptive statistical analyses were conducted to describe the demographic characteristics of adolescents. All data are reported as numbers and percentages for categorical data and mean ± SD for continuous data. Medians with interquartile ranges (IQRs) were reported for the skewed data values, Mann Whitney U test was performed for comparison in scores on different domains of EPII) between groups. Logistic regression analyses (enter method) were employed to predict demographic variables i.e., gender, residential area, family system and age groups as it allows to test models to predict binary categorical outcomes. The level of significances was set at $p < 0.05$ for all statistical analyses.

## Results

Results revealed majority of the participants (80.4%) belonged to urban areas while 48% of the participants reported that they were living with joint family system. Majority's (84%) family monthly income was reported to be less than 50,000 PKR (US$ = 326.88). only 55 (6.5%) whereas 45 (5.3%) adolescents owned up having history of psychiatric or medical problems. The demographics and general information are summarized in "Table 1".

   The frequencies of endorsements of items on Epidemic Pandemic Impact Inventory-Adolescent Domains by response option are presented in "Tables 2 and 3". Education and training were the most adversely affected domain due to closure of educational institutions and difficulties in virtual learning. Changes in responsibilities and chores at home (62%), Conflicts with parents/ other adults (44%), separation from friends (46%), cancellations of celebrations (51%) were other important impacts identified by adolescents. Home confinement and closure of schools also led to increase in screen time (52.7%), less physical activity (46.3%) and unhealthy eating habits (35.5%). "Table 2"

   In response to questions about positive impact of Pandemic, adolescents described spending more quality time with family members, siblings, having more time doing enjoyable activities, developing new hobbies, volunteering time to helping people in need. (Table 3).

   The median (Inter Quartile Range scores) for all domains of EPII-A are presented in Table 4. Overall, male adolescents consistently had significant negative impact in most EPII-A domains in comparison to females (P-value < .05). Compared with adolescents aged up to 15 years, 15–18 years old respondents reported statistically significant higher median scores on

**Table 1. Participant's sociodemographic characteristics (N = 842).**

| Variable | N (%) |
|---|---|
| **Age (Mean + SD)** | 17.14(1.48) |
| **Gender** | |
| Males | 135 (16) |
| Females | 707(84) |
| **Residence** | |
| Urban | 677(80.4) |
| Rural | 165(19.6) |
| **Education** | |
| Under matric | 69(8.2) |
| Matric | 97(11.5) |
| Higher secondary school | 328(38.9) |
| Graduation | 323(38.3) |
| Not specified | 25(3) |
| **Number of siblings** | Median = 4; Min-Max = 0–9 |
| **Birth order** | |
| 1st Child | 272(32.3) |
| Middle Child | 384(45.7) |
| Last Child | 175(20.8) |
| Only Child | 11(1.3) |
| **Family system** | |
| Nuclear Family System | 389(46.1) |
| Joint Family System | 404(48) |
| Extended Family System | 49(5.8) |
| **Family status** | |
| Intact family | 762(90.5) |
| Single parent (through death) | 66(7.8) |
| Single parent (through divorce) | 14(1.7) |
| **Family monthly income** | |
| 0–50000 | 708(84.1) |
| 50001–100000 | 86(10.2) |
| 100001–150000 | 9(1.1) |
| 150001–200000 | 11(1.3) |
| Above 200000 | 28(3.3) |
| **Father's education** | |
| Under matric | 108(12.7) |
| Matric | 337(40) |
| Higher secondary education | 172(20.4) |
| Graduation | 167(19.8) |
| Post-graduation | 58(6.9) |
| **Mother's education** | |
| Under matric | 138(16.4) |
| Matric | 353(41.9) |
| Higher secondary education | 153(18.1) |
| Graduation | 139(16.5) |
| Post-graduation | 59(7.0) |
| **Mother's occupation** | |
| House wife | 751(89.2) |

(*Continued*)

**Table 1.** (Continued)

| Variable | N (%) |
|---|---|
| Working woman | 88(10.4) |
| Missing data. | 3(0.4) |
| **Participants with chronic medical problem** | |
| Yes | 45(5.3) |
| No | 723(85.9) |
| Missing answer | 74(8.8) |
| **Participants with mental health problem in past** | |
| Yes | 55(6.5) |
| No | 713(84.7) |
| Missing Data | 74(8.8) |

social activities physical distance and quarantine and economics domains. Only a few domains were significantly affected by urban/ rural areas "Table 4". Adolescents having past medical illness and past psychiatric illness had higher scores in infection history (P value < .001). Adolescents with history of psychiatric illness had significantly higher median scores in emotional health and well-being, physical health problems and infection history domains (4.60 + 3.60 vs 2.94 + 3.39) but it was not statistically significant.

Logistic regression revealed that the domains i.e., work and employment, emotional health and well-being and infection history emerged as significant predictor for males, whereas home life and economic conditions emerged as significant predictor for females. (Table 5) For residential area, only two domains i.e., home life and emotional health and well-being emerged as significant predictors for rural population. For the outcome i.e., family system, two domains of EPII-A i.e., education and training and home life had significant impact on nuclear family system whereas the domain i.e., positive change emerged as inverse significant predictor for the outcome joint family system. For age groups, only two domains i.e., economic and infection history emerged as significant predictors for 15–18 years old.

## Discussion

Although COVID-19 has impacted adolescents' lives significantly, yet there is limited research highlighting adolescents' subjective experiences with the Pandemic and its implications. To the best of our knowledge, this study is the first to investigate the consequences of the COVID-19 pandemic on adolescents in Pakistan. This study highlights several important findings. First, the results revealed significant impact of COVID-19 Pandemic in almost all dimensions of adolescent's lives. Second, male adolescents appeared to be more adversely affected. Third, having past psychiatric history is likely to increase the vulnerability for emotional health problems during the Pandemic.

The current results suggesting that key features of this pandemic i.e., school closures, social-distancing, and the economic fallout present unique problems for adolescents, are consistent with the literature.[12] A significant proportion of adolescents' and young adults' life is in their educational activities. COVID-19 has affected the educational lives of adolescents which led to the disruption of their sleep-wake cycle associated with going to school as well as their co-curricular and extra-curricular activities in educational institutes. Regular school activities help bring structure and routine to their daily lives and shape the overall personalities of adolescents and young adults [5, 12]. More than half of the participants (50.3–64.3%)

**Table 2. Frequency & percentage endorsement on Epidemic Pandemic Impact Inventory-Adolescent domains items.**

| | | No | Yes | Not applicableN (%) |
|---|---|---|---|---|
| | | N (%) | N (%) | N (%) |
| | **WORK AND EMPLOYMENT** | | | |
| 1. | Laid off from job or could no longer work. | 324(38.4) | 211(25.0) | 307(36.4) |
| 2. | Reduced work hours. | 267(31.7) | 320(38.0) | 255(30.2) |
| 3. | Unable to start new job. | 316(37.5) | 233(27.6) | 293(34.8) |
| 4. | Had to continue to work even though in close contact with people who might be infected (for example, customers, patients, co-workers). | 344(40.8) | 222(26.3) | 276(32.7) |
| 5. | Spend a lot of time disinfecting at home due to close contact with people who might be infected at work. | 301(35.7) | 304(36.1) | 237(28.1) |
| 6. | Increase in workload or work responsibilities. | 293(34.8) | 352(41.8) | 197(23.4) |
| 7. | Parent laid off or could no longer work. | 395(46.9) | 277(32.9) | 170(20.2) |
| 8. | Parent had reduced work hours. | 364(43.2) | 334(39.6) | 144 (17.1) |
| 9. | Parent had to continue to work even though in close contact with people who might be infected (for example, customers, patients, co-workers). | 369(43.8) | 291(34.5) | 182(21.6) |
| 10 | Parent had to spend a lot of time disinfecting at home due to close contact with people who might be infected at work. | 365(43.3) | 312(37.0) | 165(19.6) |
| 11. | Parent had to increase workload or work responsibilities. | 302(35.8) | 401(47.6) | 139(16.5) |
| | **EDUCATION AND TRAINING** | | | |
| 12. | School closed or was unable to go to school. | 273(32.4) | 518(61.5) | 51(6.1) |
| 13. | Hard time participating in virtual or distance learning from home. | 279(33.1) | 519(61.6) | 44(5.2) |
| 14. | Hard time keeping up with schoolwork. | 265(31.4) | 542(64.3) | 35(4.5) |
| 15. | Unable to attend important school events (for example, prom, graduation, senior trips, dances). | 309(36.7) | 456(54.1) | 77(9.1) |
| 16. | Unable to attend afterschool activities (for example, groups, clubs, organizations). | 300(35.6) | 461(54.7) | 81(9.6) |
| 17. | Unable to participate in school athletics (for example, training, games, sports banquets). | 307(36.4) | 447(53.0) | 88(10.4) |
| 18. | Unable to participate in community or non-school related clubs and organizations. | 292(34.6) | 424(50.3) | 126(14.9) |
| 19. | Unable to complete important life steps (for example, getting driver's permit or license, visiting college or trade school, moving to college or away from home). | 316(37.5) | 404(47.9) | 122(14.5) |
| 20. | Returned home from college, boarding school, study abroad, or other away-from home living situation. | 344(40.8) | 351(41.6) | 147(17.4) |
| | **HOME LIFE** | | | |
| 21. | Difficulty taking care of siblings or other children in the home. | 375(44.5) | 403(47.8) | 64(7.6) |
| 22. | Had to spend time teaching or helping a sibling do schoolwork. | 233(27.6) | 555(65.8) | 54(6.4) |
| 23. | Changes in responsibilities or chores at home. | 259(30.7) | 523(62.0) | 60(7.1) |
| 24. | More conflict with parent(s) or other adults who look after me. | 389(46.1) | 373(44.2) | 80(9.5) |
| 25. | More conflict with siblings or other family members. | 392(46.5) | 382(45.3) | 68(8.1) |
| 26. | Limited privacy or alone time. | 381(45.2) | 403(47.8) | 58(6.9) |
| 27. | Family or friends had to move into my home. | 443(52.6) | 327(38.8) | 72(8.5) |
| 28. | Had to spend a lot more time taking care of an adult family member. | 364(43.2) | 412(48.9) | 66(7.8) |
| 29. | Had to move or relocate. | 551(65.4) | 195(23.1) | 96(11.4) |
| 30. | Became homeless. | 610(72.4) | 131(15.5) | 101(12.0) |
| | **SOCIAL ACTIVITIES** | | | |
| 31. | Separated from family or family member. | 640(75.9) | 157(18.6) | 45(5.3) |
| 32. | Separated from friend(s). | 422(50.1) | 385(45.7) | 35(4.2) |
| 33. | Separated from a girlfriend/boyfriend or romantic partner. | 340(40.3) | 187(22.2) | 315(37.4) |
| 34. | Had more arguments or conflicts with friends. | 507(60.1) | 220(26.1) | 115(13.6) |
| 35. | Had more arguments or conflict with a girlfriend/boyfriend or romantic partner. | 429(50.9) | 123(14.6) | 290(34.4) |
| 36. | Increased bullying or harassment on phone texts or social media. | 515(61.1) | 190(22.5) | 137(16.3) |
| 37. | Broke-up with a girlfriend/boyfriend or romantic partner. | 457(54.2) | 111(13.2) | 274(32.5) |
| 38. | Did not have the ability or resources to talk to family, friends, or a girlfriend/boyfriend or romantic partner while separated. | 496(58.8) | 139(16.5) | 207(24.6) |

*(Continued)*

**Table 2.** (Continued)

| | | No | Yes | Not applicableN (%) |
|---|---|---|---|---|
| | | N (%) | N (%) | N (%) |
| 39. | Unable to visit a loved one in a care facility (for example, nursing home, group home). | 451(53.5) | 245(29.1) | 146(17.3) |
| 40. | Family celebrations cancelled or restricted (for example, birthday parties, reunions). | 346(41.0) | 430(51.0) | 66(7.8) |
| 41. | Planned travel or vacations cancelled. | 330(39.1) | 430(51.0) | 82(9.7) |
| 42. | Religious or spiritual activities cancelled or restricted. | 412(48.9) | 347(41.2) | 83(9.8) |
| 43. | Unable to be with a close family member in critical condition. | 445(52.8) | 313(37.1) | 84(10.0) |
| 44. | Unable to attend in-person funeral or religious services for a family member or friend who died. | 459(54.4) | 286(33.9) | 97(11.5) |
| 45. | Unable to participate in social clubs, sports teams, or usual volunteer activities. | 347(41.2) | 386(45.8) | 109(12.9) |
| 46. | Unable to do enjoyable activities or hobbies. | 369(43.8) | 404(47.9) | 69(8.2) |
| | **ECONOMICS** | | | |
| 47. | Unable to get enough food or healthy food. | 515(61.1) | 280(33.2) | 47(5.6) |
| 48. | Unable to access clean water. | 559(66.3) | 246(29.2) | 37(4.4) |
| 49. | Unable to pay important bills like gas, car insurance, or phone bill. | 470(55.8) | 300(35.6) | 72(8.5) |
| 50. | Had trouble getting places due to less access to public transportation or concerns about safety. | 354(42.0) | 391(46.4) | 97(11.5) |
| 51. | Unable to get needed medications (for example, prescriptions or over-the-counter). | 491(58.2) | 250(29.7) | 101(12.0) |
| | **EMOTIONAL HEALTH AND WELL-BEING** | | | |
| 52. | Got into trouble more often. | 471(55.9) | 295(35.0) | 76(9.0) |
| 53. | Had increased sleep difficulties, poor sleep quality, or nightmares. | 459(54.4) | 330(39.1) | 53(6.3) |
| 54. | Had increased mental health problems or symptoms (for example, mood, anxiety, stress). | 476(56.5) | 289(34.3) | 77(9.1) |
| 55. | Used more alcohol, tobacco, vaping, or other substances. | 602(71.4) | 87(10.3) | 153(18.1) |
| 56. | Unable to access mental health treatment or therapy. | 510(60.5) | 169(20.0) | 163(19.3) |
| 57. | Not satisfied with changes in mental health treatment or therapy. | 475(56.3) | 179(21.2) | 188(22.3) |
| 58. | Spent more time on screens and devices (for example, looking at phone, playing video games, watching TV). | 339(40.2) | 444(52.7) | 59(7.0) |
| 59. | Parent had increased mental health problems or symptoms (for example, mood, anxiety, stress). | 509(60.4) | 246(29.2) | 87(10.3) |
| 60 | Parent increased use of alcohol or substances. | 590(70.0) | 95(11.3) | 157(18.6) |
| 61 | Parent unable to access mental health treatment or therapy. | 549(65.1) | 142(16.8) | 151(17.9) |
| | **PHYSICAL HEALTH PROBLEMS** | | | |
| 62. | Increased health problems not related to this disease. | 525(62.3) | 217(25.7) | 100(11.9) |
| 63. | Less physical activity or exercise. | 390(46.3) | 390(46.3) | 62(7.4) |
| 64. | Overate or ate more unhealthy foods (for example, junk food). | 478(56.7) | 299(35.5) | 65(7.7) |
| 65. | Spent more time sitting down or being sedentary. | 405(48.0) | 343(40.7) | 94(11.2) |
| 66. | Important medical procedure cancelled (for example, surgery). | 541(64.2) | 149(17.7) | 152(18.0) |
| 67. | Unable to access medical care for a serious condition (for example, dialysis, chemotherapy). | 522(61.9) | 149(17.1) | 171(20.3) |
| 68. | Got less medical care than usual (for example, routine or preventive care appointments). | 520(61.7) | 187(22.2) | 135(16.0) |
| 69. | Elderly or disabled family member not in the home unable to get the help they need. | 511(60.6) | 162(19.2) | 169(20.0) |
| 70. | Parent(s) had increased health problems not related to this disease. | 511(60.6) | 162(19.2) | 169(20.0) |
| 71. | Parent(s) important medical procedures were cancelled. | 537(63.7) | 174(20.6) | 131(15.5) |
| 72. | Parent(s) unable to access medical care for a serious condition (for example, dialysis, chemotherapy). | 544(64.5) | 123(14.6) | 175(20.8) |
| 73. | Parent(s) got less medical care than usual (for example, routine or preventive care appointments. | 519(61.6) | 180(21.4) | 143(17.0) |
| | **PHYSICAL DISTANCE AND QUARANTINE** | | | |
| 75. | Isolated or quarantined due to possible exposure to this disease. | 432(51.2) | 283(33.6) | 127(15.1) |
| 76. | Isolated or quarantined due to symptoms of this disease. | 526(62.4) | 186(22.1) | 130(15.4) |
| 77. | Isolated due to existing health conditions that increase risk of infection or disease. | 502(59.5) | 221(26.2) | 119(14.1) |
| 78. | Had limited physical closeness with a parent or loved one due to concerns of infection. | 454(53.9) | 264(31.3) | 124(14.7) |
| 79. | A close family member not in the home was quarantined. | 514(61.0) | 205(24.3) | 123(14.6) |
| 80. | A family member was unable to return home due to quarantine or travel restrictions. | 519(61.6) | 187(22.2) | 136(16.1) |

(*Continued*)

**Table 2.** (Continued)

| | | No | Yes | Not applicableN (%) |
|---|---|---|---|---|
| | | N (%) | N (%) | N (%) |
| 81. | Entire household was quarantined for a week or longer. | 489(58.0) | 224(26.6) | 129(15.3) |
| | **INFECTION HISTORY** | | | |
| 82. | Currently have symptoms of this disease but have not been tested. | 603(71.5) | 96(11.4) | 143(17.0) |
| 83. | Was tested and currently have this disease. | 632(75.0) | 85(10.1) | 125(14.8) |
| 84. | Tested positive for this disease but no longer have it. | 609(72.2) | 80(9.5) | 153(18.1) |
| 85. | Got medical treatment due to severe symptoms of this disease. | 602(71.4) | 97(11.5) | 143(17.0) |
| 86. | Had to stay in the hospital due to this disease. | 634(75.2) | 70(8.3) | 138(16.4) |
| 87. | Someone died of this disease while in our home. | 624(74.0) | 94(11.2) | 124(14.7) |
| 88. | Death of close friend or family member from this disease. | 594(70.5) | 132(15.7) | 116(13.8) |
| 89. | Parent(s) had symptoms of this disease but have not been tested. | 624(74.0) | 102(12.1) | 116(13.8) |
| 90. | Parent(s) tested and currently has this disease. | 637(75.6) | 74(8.8) | 131(15.5) |
| 91. | Parent(s) tested positive for this disease but no longer has it. | 608(72.1) | 104(12.3) | 130(15.4) |
| 92. | Parent(s) got medical treatment due to severe symptoms of this disease. | 616(73.1) | 89(10.6) | 137(16.3) |
| 93. | Parent(s) had to stay in the hospital due to this disease. | 647(76.7) | 69(8.2) | 126(14.9) |
| 94. | Someone in my family had symptoms of this disease but was never tested. | 611(72.5) | 106(12.6) | 125(14.8) |

**Table 3.** Frequency & percentage endorsement on Epidemic Pandemic Impact Inventory-Adolescent positive change domains items.

| | | No | Yes | Not applicableN (%) |
|---|---|---|---|---|
| | | N (%) | N (%) | N (%) |
| | **POSITIVE CHANGE** | | | |
| 1. | More quality time with family, friends, or romantic partner in person or from a distance (for example, on the phone, Email, social media, video conferencing, online gaming). | 298(35.3) | 449(53.3) | 95(11.3) |
| 2. | More quality time with parent(s) or other adults who look after me at home. | 239(28.4) | 526(62.4) | 77(9.1) |
| 3. | More quality time with siblings and other family members. | 259(30.7) | 521(61.8) | 62(7.4) |
| 4. | Improved relationships with family, friends, or a romantic partner. | 287(34.0) | 440(52.2) | 115(13.6) |
| 5. | New connections made with supportive people. | 343(40.7) | 397(47.1) | 102(12.1) |
| 6. | Spent more time playing and caring for pet(s). | 441(52.3) | 300(35.6) | 101(12.0) |
| 7. | Increase in exercise or physical activity. | 437(51.8) | 336(39.9) | 69(8.2) |
| 8. | More time in nature or being outdoors. | 506(60.0) | 257(30.5) | 79(9.4) |
| 9. | More time doing enjoyable activities (for example, reading books, puzzles, playing games). | 301(35.7) | 476(56.5) | 65(7.7) |
| 10. | Developed new hobbies or activities. | 342(40.6) | 435(51.6) | 65(7.7) |
| 11. | More appreciative of things usually taken for granted. | 372(44.1) | 359(42.6) | 111(13.2) |
| 12. | Paid more attention to personal health. | 331(39.3) | 433(51.4) | 78(9.3) |
| 13. | Paid more attention to preventing physical injuries. | 383(45.4) | 351(41.6) | 108(12.8) |
| 14. | Ate healthier foods. | 264(31.3) | 500(59.3) | 78(9.3) |
| 15. | Less use of alcohol, tobacco, vaping, or other substances. | 265(31.4) | 261(31.0) | 316(37.5) |
| 16. | Spent less time on screens or devices outside of work hours (for example, looking at phone, playing video games, watching TV). | 397(47.1) | 343(40.7) | 102(12.1) |
| 17. | Volunteered time to help people in need. | 344(40.8) | 377(44.7) | 121(14.4) |
| 18. | Donated time or goods to a cause related to this disease (for example, made masks, donated blood, volunteered). | 441(52.3) | 285(33.8) | 116(13.8) |
| 19. | Found greater meaning in work or school. | 362(42.9) | 386(45.8) | 94(11.2) |
| 20. | More efficient or productive in work or school. | 354(42.0) | 391(46.4) | 97(11.5) |

**Table 4. Epidemic Pandemic Impact Inventory-Adolescent domains total scores in all respondents and subgroups.** (N = 842).

| EPII-A Subscales | Total Score Median (IQR) | Gender Median (IQR) | | | Area Median (IQR) | | | Age groups Median (IQR) | | | Past Psychiatric Illness Median (IQR) | | |
|---|---|---|---|---|---|---|---|---|---|---|---|---|---|
| | | Women (n = 707) | Men (n = 135) | P. value | Urban (n = 677) | Rural (n = 165) | P. value | Age <15 years (n = 108) | Age 15-18 years (n = 734) | P. value | No Psychiatric illness (n = 797) | Psychiatric illness present (n = 45) | P. value |
| Work & Employment | 3.0 (1.0–6.2) | 3.0 (0.0–6.0) | 5.0 (2.0–8.0) | .000** | 6.0 (3.0–8.0) | 7.0 (4.0–9.0) | .034* | 2.0 (1.2–5.0) | 4.0 (0.0–7.0) | .070 | 3.0 (0.75–6.0) | 4.0 (2.0–7.0) | .291 |
| Education & Training | 5.0 (3.0–7.0) | 4.5 (2.0–7.0) | 7.0 (4.0–9.0) | .000** | 5.0 (3.0–7.0) | 4.0 (2.0–7.0) | .431 | 5.0 (3.0–6.0) | 5.0 (3.0–8.0) | .131 | 5.0 (3.0–7.0) | 6.0 (3.0–8.0) | .466 |
| Home life | 5.0 (2.0–6.0) | 4.0 (2.0–6.0) | 5.0 (3.0–6.0) | .029* | 4.0 (2.0–6.0) | 5.0 (3.0–7.0) | .000** | 4.0 (3.0–5.0) | 5.0 (2.0–6.2) | .111 | 5.0 (2.0–6.0) | 4.0 (3.0–6.0) | .965 |
| Social Activities | 5.0 (2.0–8.0) | 4.0 (1.0–8.0) | 6.0 (5.0–9.0) | .000** | 5.0 (1.0–8.0) | 4.0 (1.0–8.5) | .215 | 4.0 (2.0–6.0) | 5.0 (2.0–8.0) | .029* | 5.0 (2.0–8.0) | 6.0 (3.0–10.0) | .590 |
| Economics | 1.0 (0.0–3.0) | 1.0 (0.0–3.0) | 1.0 (0.0–3.0) | .590 | 1.0 (0.0–3.0) | 2.0 (0.0–2.0) | .065 | 1.0 (0.0–2.0) | 1.0 (0.0–3.0) | .001** | 1.0 (0.0–3.0) | 1.0 (0.0–32.0) | .732 |
| Emotional health & Well being | 2.0 (0–4.0) | 2.0 (0–4.0) | 4.0 (1.75–7.0) | .000** | 2.0 (0.0–4.0) | 2.0 (0.0–5.0) | .886 | 1.5 (1.0–3.0) | 2.0 (0.0–4.0) | .207 | 2.0 (0.0–4.0) | 4.0 (1.0–6.0) | .000** |
| Physical health problems | 2.0 (0.0–4.0) | 1.0 (0.0–4.0) | 4.0 (2.0–9.0) | .000** | 2.0 (0.0–4.0) | 2.0 (0.0–5.0) | .136 | 2.0 (1.0–3.7) | 2.0 (0.0–5.0) | .979 | 2.0 (0.0–9.0) | 4.0 (2.0–6.0) | .000** |
| Physical distance & Quarantine | 1.0 (0.0–3.0) | 0.0 (0.0–3.0) | 2.0 (1.0–5.0) | .000** | 1.0 (0.0–3.0) | 1.0 (0.0–3.0) | .154 | 1.0 (0.0–2.0) | 1.0 (0.0–3.2) | .50* | 1.0 (0.0–3.0) | 2.0 (0.0–5.0) | .025 |
| Infection History | 0.0 (0.0–2.0) | 0.0 (0–1.0) | 3.0 (3.0–5.0) | .000** | 0.0 (0–2.0) | 0.0 (0–1.0) | .942 | 0.0 (0.0–2.0) | 0.0 (0.0–1.0) | .059 | 0.0 (0–1.0) | 1.0 (0.0–5.0) | .000** |
| Positive Impact | 9.50 (4.0–14.0) | 9.0 (3.75–14.0) | 8.0 (4.0–11.0) | .348 | 9.0 (4–13.0) | 10.0 (3–16.0) | .377 | 8.0 (6–11.0) | 10.0 (4–14.0) | .146 | 10.0 (4–14.0) | 9.0 (3.00–12.00) | .502 |

Abbreviations: IQR, Interquartile Range.

**P value < .001,

*P value < .05

**Table 5. Logistic regression analyses for impact of gender, residential area, family system and age on different domains of EPII-A.**

| Variables | β | OR (95% CI) | p | β | OR (95% CI) | p | β | OR (95% CI) | p | β | OR (95% CI) | p |
|---|---|---|---|---|---|---|---|---|---|---|---|---|---|
| Work & Employment | -.86** | 11.34 (.79-.94) | .001 | .96 | 1.26 (.89–1.03) | .261 | .97 | .03 (.95–1.06) | .871 | .98 | .097 (.91–1.07) | .756 |
| Education & Training | .96 | 1.35 (.89–1.03) | .246 | .99 | .73 (.96–1.09) | .393 | .99* | .07 (1.01–1.62) | .014 | .99 | .34 (.95–1.11) | .556 |
| Home Life | .14** | 1.15 (1.03–1.27) | .008 | .13** | 1.14 (1.05–1.24) | .002 | .08* | 1.09 (1.01–1.16) | .014 | .95 | 1.02 (.86–1.04) | .310 |
| Social Activities | .96 | 1.65 (.89–1.02) | .199 | .99 | .103 (.93–1.05) | .749 | .99 | .18 (.95–1.03) | .670 | .99 | .87 (.96–1.10) | .351 |
| Economic | .22** | 1.24 (1.07–1.43) | .003 | .99 | .54 (.93–1.17) | .462 | .99 | .05 (.90–1.08) | .822 | .99** | 7.62 (1.06–1.43) | .006 |
| Physical Health | .94 | 2.26 (.87–1.02) | .133 | .98 | 2.21 (.98–1.14) | .137 | .98 | 2.59 (.98–1.12) | .107 | .98 | 1.22 (.92–1.26) | .727 |
| Physical Distance | .98 | 0.15 (.88–1.09) | .702 | .99 | .542 (.94–1.15) | .462 | .98 | .08 (.91–1.07) | .778 | .99 | 2.01 (.96–1.25) | .156 |
| Emotional Health & Well-being | -.13* | .88 (.78-.98) | .027 | .11* | .89 (.81-.99) | .027 | .97 | .42 (.89–1.05) | .517 | .99 | .06 (.89–1.05) | .800 |
| Infection History | -.13** | .88 (.83-.94) | .000 | .99 | .08 (.92–1.06) | .777 | .97 | .99 (.92–1.03) | .318 | .89** | 6.89 (.81-.97) | .009 |
| Positive Change | .05** | 1.05 (1.00–1.10) | .000 | .99 | .12 (.96–1.03) | .733 | -.04* | .97 (.94-.99) | .029 | .99 | .58 (.97–1.06) | .446 |

Note.

*$p < .05$.

**$p < .01$.

Gender: $R^2$ = .13 (Cox & Snell),.21 (Nagelkerke). Model Chi-Square = 20.20 p < .01, Residential Area: $R^2$ = .03 (Cox & Snell),.04 (Nagelkerke). Model Chi-Square = 22.64 p < .01, Family System: $R^2$ = .02 (Cox & Snell),.03 (Nagelkerke). Model Chi-Square = 2.64 p = .95 and Age: $R^2$ = .03 (Cox & Snell),.05 (Nagelkerke). Model Chi-Square = 6.44 p = .598.

reported difficulties in schoolwork and disruption of their co-curricular, extra-curricular activities and important milestones like graduation and educational trips. Difficulty in schoolwork can be attributed to the sudden change in the medium of delivery [2]. Unfamiliarity with the virtual learning process can impact their learning abilities and lead to increased difficulty with schoolwork. Gender was significantly associated with the impact on education and training. This is concordant with the study by Laar et al, which shows that the female students are less active in participation in the co-curricular and extra-curricular activities due to perceived socio-cultural and religious limitation, which may be why they are less affected by the impact of COVID-19 on these activities [17].

The majority of the workplaces have been closed due to the COVID-19 pandemic. The economic crisis further potentiates negative downstream effects on adolescents through possible impact on any part time work they were involved in to supplement family income as well as parent unemployment, parent mental health, and household conflicts [18]. Qualitative research in India reported the financial losses, loss of father's job, and unavailability of daily needs as the major sources of stress in adolescents during COVID-19 [6]. This is concordant with our study showing that approximately 1/3rd (29.2–46.4%) of the participants had economic difficulties including access to daily needs like clean water, food, transportation, bills, and medications.

Another important aspect to be considered during this pandemic is the quality of life that the adolescents are spending at their homes, as the lockdowns were implemented globally [4]. Almost half of the participants (47.8–65.8%) reported decreased quality of home life, including difficulty with taking care of siblings or family members, increase in responsibilities, conflicts with parents and family members, and limited privacy. A study by J Zhou et al in China shows that 14.1% of children spending more than 10 hours with their parents reported an increase in conflicts with their parents [1]. The prevalence in our study was much higher (44.2%). Impact on home life was also found to be significantly associated with gender and area of residence. Thus, it can be postulated that in a male-dominant society like Pakistan where males have more freedom to go out and thus more available privacy, and fewer responsibilities at home,

the effect of staying at home was greater for males. Social activities have also been halted as a result of the COVID-19 pandemic. Important gatherings like birthday celebrations, reunions, planned vacations, religious activities like Friday/Eid prayer, and sports were canceled for 1/3rd to half of the participants. Moreover, adolescents (45%) reported separation from friends as one of the factors that impact their social life. Social support is also an important buffer that helps stress in adolescents and young adults. A study by Baloch GM et al suggests that female university students were more pro-active in using social support and humanitarian strategies as coping skills compared to their male counterparts during COVID-19 [18]. This may be one of the reasons for the noteworthy effect of social life disruption in boys.

The results extended previous findings by demonstrating that the effect of COVID-19 on multiple aspects of the lives of adolescents has resulted in increased psychological morbidity as well [10, 19]. Sleep difficulties were the second most reported emotional health problem (39.1%), which is due to the disruption of structured life and routine as well the increased screen use [12]. A striking 52.7% reported increased screen time which is congruent with the previously reported data during COVID-19 [10, 11]. Increased internet use can also lead to worsening of mental health, which may be even more significant in adolescents with pre-existing psychiatric problems [8, 15]. This is in agreement with our results where past psychiatric illness and gender were significantly associated with emotional health during COVID-19. Quarantine and exposure to COVID-19 were also major contributing factors to the impact of COVID-19 on adolescents. It is interesting to note that 62.4% of the participants were isolated or quarantined which was significantly associated with gender and past psychiatric illness. As mentioned earlier, male adolescents may feel a loss of control over their freedom and social life while being in quarantine. Likewise, adolescents with past psychiatric problems are more susceptible to the impact of social distancing and quarantine [4, 5, 15].

The physical health of adolescents also suffered a lot during COVID-19. Multiple studies reported a weight gain in adolescents during the lockdown [12, 13]. Our study also reported that 35.5%-46.3% of the participants were involved in decreased physical activity, adopted a sedentary lifestyle, and had an increased intake of unhealthy foods. Our findings resonated with the results of previous studies [8, 9, 14]. Boys in Pakistan in particular mostly rely on out-of-home sports for physical activities, which are significantly disrupted during COVID-19. Similarly, participants with previous physical or mental health problems are more vulnerable to changes in their environment and their health may be significantly impacted by changes in physical activity, diet changes, or closure of elective hospital facilities [8].

Meanwhile, COVID-19 also brought a positive change in the lives of adolescents. Zhou et al argued that disasters like the COVID-19 pandemic bring challenges but also offer opportunities for adolescents to nurture [1]. 45.6% of adolescents in their study showed post-traumatic growth which is similar to our findings in which 30.5–62.4% of adolescents showed a positive change in their lives. Nan Zho reported that in collectivistic societies like ours, social adaptation to problems may help adolescents develop a better sense of control and healthy coping strategies to master these challenges [20]. Adolescents reported much greater time spent with parents, a trend that our results suggested was generally taken positively.

Nevertheless, there are several limitations to the current study. First, the cross-sectional study design precluded the ability to establish a causal relationship. Second, as the online survey method rely on the self-selection of respondents and all data was self-reported, may lead to biased estimates. Finally, majority of participants were females and from urban areas of one city thus findings may not be applicable to adolescents residing in other regions. Future work may wish to use a multimethod approach for assessing the long- term impact of Pandemic.

Despite these limitations, our results confirmed significant impact and various challenges faced by adolescents during the Pandemic in Pakistan, which was the main aim of the study.

COVID-19 has a marked impact on the intra-individual, inter-individual as well as environmental dimensions of the life of adolescents including education, home and social life, mental and physical health. Whether the impact is positive or negative, its significance on the overall shaping of adolescents' lives cannot be overlooked. Boys, older age adolescents and adolescents with past history of Psychiatric illness appear to be more vulnerable. Adolescents experience both post-traumatic stress and growth consequent to a trauma like COVID-19, thus it is important to devise ways to alleviate the stress and enhance the coping strategies of the adolescents to help them effectively cope with this menace in all aspects of their lives. Particular attention needs to be paid to boys, older adolescents and those with previous history of psychiatric illness. Our findings inform policy makers, school officials and parents, of adolescents own experiences highlighting that efforts should prioritize this vulnerable group, as pandemic impacts continue to evolve.

## Supporting information

**S1 File. Epidemic Pandemic Impact Inventory-Adolescent domains total mean scores of participants and subgroups.**
(PDF)

## Acknowledgments

The authors are grateful to the participants of different public institutions who spare their time and participated in the research. We are also grateful for Dr Damion Grasso and his team for permission to use EPII Adolescent supplement.

## Author Contributions

**Conceptualization:** Nazish Imran, Fauzia Naz, Muhammad Waqar Azeem.

**Data curation:** Nazish Imran, Fauzia Naz, Muhammad Imran Sharif, Sumbul Liaqat, Musarrat Riaz, Abida Khawar.

**Formal analysis:** Nazish Imran, Fauzia Naz, Muhammad Imran Sharif, Sumbul Liaqat, Musarrat Riaz.

**Investigation:** Musarrat Riaz.

**Methodology:** Nazish Imran, Fauzia Naz, Muhammad Imran Sharif, Sumbul Liaqat, Musarrat Riaz, Abida Khawar, Muhammad Waqar Azeem.

**Project administration:** Nazish Imran, Muhammad Imran Sharif, Abida Khawar.

**Supervision:** Muhammad Waqar Azeem.

**Writing – original draft:** Nazish Imran, Fauzia Naz, Muhammad Imran Sharif, Sumbul Liaqat, Musarrat Riaz, Muhammad Waqar Azeem.

**Writing – review & editing:** Nazish Imran, Fauzia Naz, Muhammad Imran Sharif, Sumbul Liaqat, Musarrat Riaz, Abida Khawar, Muhammad Waqar Azeem.

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
