## [Decision Letter · Decision Letter 0]

16 Sep 2021

PONE-D-21-23130Multidimensional impacts of coronavirus pandemic in adolescents in Pakistan: A cross sectional researchPLOS ONE

Dear Dr. Imran,

Thank you for submitting your manuscript to PLOS ONE. After careful consideration, we feel that it has merit but does not fully meet PLOS ONE’s publication criteria as it currently stands. Therefore, we invite you to submit a revised version of the manuscript that addresses the points raised during the review process.

ACADEMIC EDITOR: Considering the reviewers comments and my own reading of the paper, I am suggesting a major revision for this paper. Please carefully address and respond to some serious queries put fort by reviewer 3. The revision will not give guarantee unless the reviewers are happy with your revisions or responses.  

We look forward to receiving your revised manuscript.

Kind regards,

Srinivas Goli, Ph.D.

Academic Editor

PLOS ONE

Additional Editor Comments (if provided):

Considering the reviewers comments and my own reading of the paper, I am suggesting a major revision for this paper. Please carefully address and respond to some serious queries put fort by reviewer 3. The revision will not give guarantee unless the reviewers are happy with your revisions or responses.

2. Please provide additional details regarding participant consent. In the Methods section, please ensure that you have specified (1) whether consent was informed and (2) what type you obtained (for instance, written or verbal). If your study included minors, state whether you obtained consent from parents or guardians. If the need for consent was waived by the ethics committee, please include this information.

Reviewers' comments:

Reviewer's Responses to Questions

**Comments to the Author**

1. Is the manuscript technically sound, and do the data support the conclusions?

Reviewer #1: No

Reviewer #2: Partly

Reviewer #3: Partly

2. Has the statistical analysis been performed appropriately and rigorously? 

Reviewer #1: No

Reviewer #2: No

Reviewer #3: No

3. Have the authors made all data underlying the findings in their manuscript fully available?

Reviewer #1: Yes

Reviewer #2: No

Reviewer #3: No

4. Is the manuscript presented in an intelligible fashion and written in standard English?

Reviewer #1: No

Reviewer #2: No

Reviewer #3: No

5. Review Comments to the Author

Reviewer #1: The research topic is good and appears to be relevant and significant during the pandemic. The research methodology raises some questions. First, the Adolescent Pandemic-Adaptation Impact Inventory (EPII-A) is intended to measure many domains. Whereas, the authors measured only some domains mentioned in the "Results section". It is recommended to justify which domains are evaluated and which domains are not evaluated. Second, the main focus of data analysis is descriptive analysis. What about measuring the impact of the pandemic on adolescents, which seemed to be the main objective of the study? Finally, table number 2-4 does not seem to be appropriate. Some grammatical errors are also visible.

Reviewer #2: Strong and relevant literature review. Sounf problem identification and back drop of research with some grammatical errors, however, significant improvement can be made in statistical analysis. Please consider the following points:

1) Sample design and considerations could be added. Sample does not seem to be proportionate to the universe. i.e, all adolescents in Pakistan (data could be found on PBS). Data is currently skewed toward lower PSLM, where child labor seems to be prevalent as several respondents have claimed to have lost their job and reduced work hour. So my question is that how is the author claiming that education of adolescents is affected based on this data, when most of them were engaged in economic activites before COVID-19. We can perhaps conclude this for anecdotal evidence

2)Sample is skewed towards one gender only. Is this intentional?

3) When using independent t test we need to make sure that that variables are independent, are normally distributed and are there is no presence of heterogeneity of variance. Please look into Kolmogorov-Smirnov (K-S) test or Anderson-Darling Test.

4) Could run simple OLS to create a link between the present literature and add on too the already published findings.

5) Correlation matrix an be used to further this analysis. Chi-square and Anova testing are also a good to go options for continuous data sets.

6) since no proper testing is done, no hypothesis is given and nothing is being proved or disproved. The author has claimed the conclusions solely on descriptive statistics from the collected data.

7) EPII-A items could be integrated in the model.

Summing up, this is more of an elaborate literature review backed by averages from the data. The required statistical depth.

Reviewer #3: 1.This study takes 13-18 years old individuals as adolescent, while WHO takes 10-19 years old individuals as adolescent (https://www.who.int/southeastasia/health-topics/adolescent-health). Authors should indicate why they have used different definition of adolescents in the study.

2.The study has some selectivity biases in the sampling, for example, more girls (84%) than boys (16%) were interviewed. Only public sector students were interviewed. Similarly, urban participants were three-fourth of the total participants interviewed. Authors should explain whether they have adjusted the sample for rural and urban population using appropriate weights.

3. In the section on methods and setting (line 150-152), authors mentioned that they had collected the data from public sector schools and college students, while results mentioned for the work-related impacts in Table 3 showed that 26% participants, presumably females, were exposed to infected people through their interactions at the workplace. Were study participants working part time before the pandemic? Authors need to specify the work nature in discussion and results. Authors should explain 4 and 6 in Table 3, what did they mean by increase in the workload for students (4, Table 3) and continue to work with infected people (6, Table 3), particularly for adolescent girls.

4. The analysis is heavily based on female respondents (84%) while culturally female adolescents are supposed to take care of younger siblings and take part in household chores in Pakistan, authors did not mention how did they segregate the COVID impact on female responsibilities towards home from the routine responsibilities.

5. According to WHO data, only 2% death (https://www.worldometers.info/coronavirus/country/pakistan/)

occurred due to COVID 19, while this study reports 15% deaths of the close ones or relatives of the study participants, the results need to be justified using a district level or area wise COVID 19 related death rates.

6. A large t-score indicates that the groups are different, and a small t-score indicates that the groups are similar. Authors should explain p-value of the mean difference test for gender in Table 4 corresponding to work and employment.

7. Authors should test the assumption of homogeneity of the variance and report results in the paper.

8. Authors need to develop a clear connection between the objective and results of the study while discussing results in the manuscript.

9. Policy implications need to be synchronized with the main research findings.

10. Conclusion should include a discussion on the motivation, study objectives, methodology, and results more explicitly.

6. PLOS authors have the option to publish the peer review history of their article (what does this mean?). If published, this will include your full peer review and any attached files.

Reviewer #1: **Yes: **Mahmooda Aftab

Reviewer #2: **Yes: **Dr. Atiya Aabroo

Reviewer #3: **Yes: **Lubna Naz

---

## [Author Response · Author response to Decision Letter 0]

31 Oct 2021

We have modified the title page and manuscript to follow the PLOS ONE’s style requirements.

2. Please provide additional details regarding participant consent. In the Methods section, please ensure that you have specified (1) whether consent was informed and (2) what type you obtained (for instance, written or verbal). If your study included minors, state whether you obtained consent from parents or guardians. If the need for consent was waived by the ethics committee, please include this information.

Additional details added in methods.

All relevant data are within the manuscript and its supporting files. 

5. Review Comments to the Author

Reviewer #1: The research topic is good and appears to be relevant and significant during the pandemic. The research methodology raises some questions. First, the Adolescent Pandemic-Adaptation Impact Inventory (EPII-A) is intended to measure many domains. Whereas, the authors measured only some domains mentioned in the "Results section". It is recommended to justify which domains are evaluated and which domains are not evaluated. Second, the main focus of data analysis is descriptive analysis. What about measuring the impact of the pandemic on adolescents, which seemed to be the main objective of the study? Finally, table number 2-4 does not seem to be appropriate. Some grammatical errors are also visible.

All the ten domains of Adolescent Pandemic-Adaptation Impact Inventory (EPII-A) were used in the present study as described in detail in Tables 2 &Table 3 of results. Due to lengthy inventory, only those statements which were endorsed by majority of respondents were highlighted in text section of results to avoid replication of results given in tables.

Tables 2 & 3 gives descriptive statistics in the way described and scored by the Inventory developers. Median and Interquartile ranges in different groups in Table 4 helps in exploring the impact of covid in various domains of their lives, which is the main objective of the study. There are no cut offs currently described by the inventory authors as it is a newly developed measure, Higher the scores, higher the impact in that specific domain. 

Grammatical errors are corrected. 

Reviewer #2: Strong and relevant literature review. Sound problem identification and back drop of research with some grammatical errors, however, significant improvement can be made in statistical analysis. Please consider the following points:

1) Sample design and considerations could be added. Sample does not seem to be proportionate to the universe. i.e, all adolescents in Pakistan (data could be found on PBS). Data is currently skewed toward lower PSLM, where child labor seems to be prevalent as several respondents have claimed to have lost their job and reduced work hour. So my question is that how is the author claiming that education of adolescents is affected based on this data, when most of them were engaged in economic activites before COVID-19. We can perhaps conclude this for anecdotal evidence

Sampling was non probability convenience sampling and therefore we agree with worthy reviewer, and have mentioned it as a limitation that sample is not representative of all adolescents in Pakistan. 

Participating schools and colleges were of public sector, which are attended mostly by low and middle social class in Pakistan as those belonging to upper middle class and elite classes prefer private schools and colleges. It is not uncommon for adolescents especially girls (majority in our sample) to work parttime along with their own studies, mostly providing tuition to children in neighborhood to supplement family income. As noted in table 3, most of work and employment statements were endorsed by around 25% of respondents in comparison to almost three fourth of participants (65% ) for education and training. That is the reason for statement in result that education and training was one of the most affected domains in current cohort.

2)Sample is skewed towards one gender only. Is this intentional?

No, It was not intentional. It may be that the participating institutions had more female students to begin with and so were more likely to participate. Also, difficult to know to what extent the length of questionnaire might have led to reluctance to participate especially from boys. No incentive was offered for participation. 

3) When using independent t test we need to make sure that that variables are independent, are normally distributed and are there is no presence of heterogeneity of variance. Please look into Kolmogorov-Smirnov (K-S) test or Anderson-Darling Test.

As per worthy reviewer suggestion and after checking the assumptions, Mann Whitney U test has been applied instead of t test. Tables and results modified accordingly.

4) Could run simple OLS to create a link between the present literature and add on too the already published findings.

Regression done and added to results.

5) Correlation matrix an be used to further this analysis. Chi-square and Anova testing are also a good to go options for continuous data sets.

Correlation added.

6) since no proper testing is done, no hypothesis is given and nothing is being proved or disproved. The author has claimed the conclusions solely on descriptive statistics from the collected data. 

It was an exploratory study and conclusions are on basis of descriptive and other statistics done as per advice of worthy reviewer.

Summing up, this is more of an elaborate literature review backed by averages from the data. The required statistical depth.

Reviewer #3: 1.This study takes 13-18 years old individuals as adolescent, while WHO takes 10-19 years old individuals as adolescent (https://www.who.int/southeastasia/health-topics/adolescent-health). Authors should indicate why they have used different definition of adolescents in the study.

We did not used different definitions of adolescents in the study. The participating institutions felt that questionnaire is too lengthy to be filled in by students aged less than 13 (corresponding to grade 6-7 in Pakistan for most adolescents). Thus parents of grade 8 and above were approached only for consent to participate, This has resulted in age range of our sample from 13-18. Also in Pakistan 18 is the legal age limit and also correspond to time when most student have completed their college intermediate education thus the upper limit of 18 noted in the sample.

2.The study has some selectivity biases in the sampling, for example, more girls (84%) than boys (16%) were interviewed. Only public sector students were interviewed. Similarly, urban participants were three-fourth of the total participants interviewed.

Selection bias has been mentioned as a limitation in discussion section (lines 331, 332 “majority of participants were females and from urban areas of one city thus findings may not be applicable to adolescents residing in other regions). Sampling was non probability convenience sampling and furthermore online survey method rely on the self-selection of respondents. It was an initial study looking at exploring impact on adolescents with plans for large scale study to have representation from rural areas as well. 

3. In the section on methods and setting (line 150-152), authors mentioned that they had collected the data from public sector schools and college students, while results mentioned for the work-related impacts in Table 3 showed that 26% participants, presumably females, were exposed to infected people through their interactions at the workplace. Were study participants working part time before the pandemic? Authors need to specify the work nature in discussion and results. Authors should explain 4 and 6 in Table 3, what did they mean by increase in the workload for students (4, Table 3) and continue to work with infected people (6, Table 3), particularly for adolescent girls.

Participating schools and colleges were of public sector, which are attended mostly by low and middle social class in Pakistan as those belonging to upper middle class and elite classes prefer private schools and colleges. It is not uncommon for adolescents especially girls (majority in our sample) to work parttime along with their own studies, mostly providing tuition to children in neighborhood to supplement family income. We did not collected specific information on type of parttime work, respondents were involved in, prior to pandemic and therefore unable to add work nature in discussion and results. 

In our observation, with school closures in Lahore, many parents (although not all) continued to send younger children to tutions despite social restrictions, for continuity of studies and help with home work sent by school particularly in instances where mother was illetrate and was not able to support child learning. Also rate of vaccinations were low at the time of study. That may account for instances where infections may have been transmitted through work. 

Increase in workload for students is understood as students having to complete work/ assignments at home on their own in many instances due to limited online learning system infrastructure in place during initial few months of Pandemic in the country. 

4. The analysis is heavily based on female respondents (84%) while culturally female adolescents are supposed to take care of younger siblings and take part in household chores in Pakistan, authors did not mention how did they segregate the COVID impact on female responsibilities towards home from the routine responsibilities.

It was not possible to segregate the COVID impact on female responsibilities towards home from the routine responsibilities in current study. This is one of the limitations mentioned in lines 329-331 of manuscript that “the online survey method rely on the self-selection of respondents and all data was self-reported, which may lead to biased estimates.”

5. According to WHO data, only 2% death (https://www.worldometers.info/coronavirus/country/pakistan/)

occurred due to COVID 19, while this study reports 15% deaths of the close ones or relatives of the study participants, the results need to be justified using a district level or area wise COVID 19 related death rates.

The study sample was from District Lahore School and colleges. Main cities including Lahore were the ones badly hurt by COVID Pandemic and reported more mortality/ morbidity compared to rural areas in Pakistan. (https://covid.gov.pk) Also under reporting of cases, reluctance of patients to go to hospital and deaths and burials without knowing exact illness was not uncommon especially in first few months of illness.due to compulsory quarantine policy, infodemic about poison injections in hospitals and taking money from International organizations for dead bodies and stigma of illness, (References)

Imran, N., Afzal, H., Aamer, I., Hashmi, A., Shabbir, B., Asif, A., & Farooq., S. (2020). Scarlett Letter: A study based on experience of stigma by COVID-19 patients in quarantine. Pakistan Journal of Medical Sciences, 36(7). https://doi.org/10.12669/pjms.36.7.3606

Battling the Infodemic- A Cross Sectional Study of General Population of Pakistan Irum Aamer , Zainab Pervaiz , Fauzia Cheema , Nazish Imran Esculapio - Volume 16, Supplement 01, Special COVID-19 Issue, 2020 - www.esculapio.pk - 66 66-72

All these factors in our opinion may have contributed towards high 15% deaths of the close ones or relatives of the study participants, 

6. A large t-score indicates that the groups are different, and a small t-score indicates that the groups are similar. Authors should explain p-value of the mean difference test for gender in Table 4 corresponding to work and employment.

Statistical tests have been applied according to the nature of data. Median and Interquartile ranges have been reported instead of t test.

7. Authors should test the assumption of homogeneity of the variance and report results in the paper.

Explained in methods that data was skewed and thus Mann whitney test is applied and median and interquartile ranges are reported.

8. Authors need to develop a clear connection between the objective and results of the study while discussing results in the manuscript.

The results are preliminary and the measure (EPIA-A) is recently developed and thus we were unable to find any published literature so far using this inventory. Therefore, Discussion focuses on covid 19 impact on various domains of adolescent’s lives noticed in current preliminary research which was the objective of the study and possible reasons for it as well as implications. It can act as baseline for further methodologically sound research on the topic. 

9. Policy implications need to be synchronized with the main research findings.

Sentence added at the end of paper highlighting vulnerable groups observed in study.

10. Conclusion should include a discussion on the motivation, study objectives, methodology, and results more explicitly. 

Last parargraph modified in line with worthy reviewer suggestion.

---

## [Decision Letter · Decision Letter 1]

9 Dec 2021

PONE-D-21-23130R1Multidimensional impacts of coronavirus pandemic in adolescents in Pakistan: A cross sectional researchPLOS ONE

Dear Dr. Imran,

Thank you for submitting your manuscript to PLOS ONE. After careful consideration, we feel that it has merit but does not fully meet PLOS ONE’s publication criteria as it currently stands. Therefore, we invite you to submit a revised version of the manuscript that addresses the points raised during the review process.

ACADEMIC EDITOR: Considering my own reading and reviewer suggestion, I am recommending minor revision before accepting this paper for publication in PLOS One.  Carefully check for the language errors and formatting issues before submitting the final version. 

We look forward to receiving your revised manuscript.

Kind regards,

Srinivas Goli, Ph.D.

Academic Editor

PLOS ONE

Journal Requirements:

Additional Editor Comments:

Considering my own reading and reviewer suggestion, I am recommending minor revision before accepting this paper for publication in PLOS One.

Reviewers' comments:

Reviewer's Responses to Questions

**Comments to the Author**

1. If the authors have adequately addressed your comments raised in a previous round of review and you feel that this manuscript is now acceptable for publication, you may indicate that here to bypass the “Comments to the Author” section, enter your conflict of interest statement in the “Confidential to Editor” section, and submit your "Accept" recommendation.

Reviewer #1: All comments have been addressed

2. Is the manuscript technically sound, and do the data support the conclusions?

Reviewer #1: Partly

3. Has the statistical analysis been performed appropriately and rigorously? 

Reviewer #1: No

4. Have the authors made all data underlying the findings in their manuscript fully available?

Reviewer #1: Yes

5. Is the manuscript presented in an intelligible fashion and written in standard English?

Reviewer #1: Yes

6. Review Comments to the Author

Reviewer #1: How was the reliability and validity of the Pandemic-Adolescent Pandemic Impact Inventory determined? What were the hypotheses of the study and What were the assumptions of using the Mann Whitney U test and Logistic Regression?

7. PLOS authors have the option to publish the peer review history of their article (what does this mean?). If published, this will include your full peer review and any attached files.

Reviewer #1: **Yes: **Mahmooda Aftab

---

## [Author Response · Author response to Decision Letter 1]

14 Dec 2021

PONE-D-21-23130R1

Multidimensional impacts of coronavirus pandemic in adolescents in Pakistan: A cross sectional research

PLOS ONE

Journal Requirements:

Additional Editor Comments:

Considering my own reading and reviewer suggestion, I am recommending minor revision before accepting this paper for publication in PLOS One.

Reviewers' comments:

Reviewer's Responses to Questions

Comments to the Author

1. If the authors have adequately addressed your comments raised in a previous round of review and you feel that this manuscript is now acceptable for publication, you may indicate that here to bypass the “Comments to the Author” section, enter your conflict of interest statement in the “Confidential to Editor” section, and submit your "Accept" recommendation.

Reviewer #1: All comments have been addressed

2. Is the manuscript technically sound, and do the data support the conclusions?

Reviewer #1: Partly

3. Has the statistical analysis been performed appropriately and rigorously?

Reviewer #1: No

4. Have the authors made all data underlying the findings in their manuscript fully available?

Reviewer #1: Yes

5. Is the manuscript presented in an intelligible fashion and written in standard English?

Reviewer #1: Yes

6. Review Comments to the Author

Reviewer #1: How was the reliability and validity of the Pandemic-Adolescent Pandemic Impact Inventory determined? What were the hypotheses of the study and What were the assumptions of using the Mann Whitney U test and Logistic Regression?

EPII-A is a newly developed assessment measure and is still in the process of determining psychometric properties. (Morris, A.S., Ratliff, E. L., Grasso, D. J., Briggs-Gowan, M. J., Ford, J. D., & Carter, A.S. (2020). The Epidemic – Pandemic Impacts Inventory Adolescent Adaptation (EPII-A). University of Connecticut School of Medicine.)The authors of EPII-A has also invited researchers world widely to employ the inventory to find empirical evidences to determine psychometric properties of the inventory in general population. Main psychometric results of the EPII-Adolescents Adaptation version i.e., construct validity, EFA, item analysis and Cronbach’s alpha reliabilities in our population revealed good properties and are being reported separately.(Lines methods 156-158)[unpublished Manuscript]. Cronbachs alpha was 0.76 for most items of EPII Adolescent adaptation.

Study was exploratory in nature and main hypothesis we had in mind was that COVID-19 Pandemic has significant impact on adolescents in all domains of life, irrespective of gender, area of residence and other variables.

As data was skewed thus Mann whitney test is applied and median and interquartile ranges were reported. OLS was done as per one of the reviewers suggestions to create a link between the present literature and add on to the already published findings. Logistic regression analyses (enter method) were employed to predict demographic variables i.e., gender, residential area, family system and age groups as it allows to test models to predict binary categorical outcomes. (mentioned in statistical analysis lines 163-167)

7. PLOS authors have the option to publish the peer review history of their article (what does this mean?). If published, this will include your full peer review and any attached files.

Do you want your identity to be public for this peer review? For information about this choice, including consent withdrawal, please see our Privacy Policy.

Reviewer #1: Yes: Mahmooda Aftab

---

## [Editor Report · Decision Letter 2]

23 Dec 2021

Multidimensional impacts of coronavirus pandemic in adolescents in Pakistan: A cross sectional research

PONE-D-21-23130R2

Dear Dr. Imran,

We’re pleased to inform you that your manuscript has been judged scientifically suitable for publication and will be formally accepted for publication once it meets all outstanding technical requirements.

Kind regards,

Srinivas Goli, Ph.D.

Academic Editor

PLOS ONE

Additional Editor Comments (optional):

Revisions are satisfactory, thus I am recommending this article for publication in PLOS One.
---

## [Editor Report · Acceptance letter]

27 Dec 2021

PONE-D-21-23130R2 

Multidimensional impacts of coronavirus pandemic in adolescents in Pakistan: A cross sectional research 

Dear Dr. Imran:

I'm pleased to inform you that your manuscript has been deemed suitable for publication in PLOS ONE. Congratulations! Your manuscript is now with our production department. 

Kind regards, 

on behalf of

Dr. Srinivas Goli 

Academic Editor

PLOS ONE